# Evaluation of a marker independent isolation method for circulating tumor cells in esophageal adenocarcinoma

Annouck Philippron[1,2,3]*, Lieven Depypere[4,5], Steffi Oeyen[6,7], Bram De Laere[3,8,9], Charlotte Vandeputte[2,3,10], Philippe Nafteux[4], Katleen De Preter[2,3,10], Piet Pattyn[1]

1 Department of Gastro-Intestinal Surgery, University Hospital Ghent, Ghent, Belgium, 2 Center for Medical Genetics, Ghent University, Ghent, Belgium, 3 Cancer Research Institute (CRIG), Ghent University, Ghent, Belgium, 4 Department of Thoracic Surgery, University Hospital Leuven, Leuven, Belgium, 5 Laboratory of Respiratory Diseases and Thoracic Surgery (BREATHE), Department of Chronic Diseases and Metabolism (CHROMETA), KULeuven, Leuven, Belgium, 6 Center for Oncological Research (CORE), Faculty of Medicine and Health Sciences, University of Antwerp, Antwerp, Belgium, 7 Translational Cancer Research Unit (TCRU), GZA Hospitals, Sint-Augustinus, Antwerp, Belgium, 8 Department of Medical Epidemiology and Biostatistics, Karolinska Institutet, Stockholm, Sweden, 9 Department of Human Structure and Repair, Ghent University, Ghent, Belgium, 10 Department of Biomolecular Medicine, Ghent University, Ghent, Belgium

* annouck.philippron@ugent.be

**Data Availability Statement:** All relevant data are within the paper and its Supporting information files.

## Abstract

### Objective

The enrichment of circulating tumor cells (CTCs) from blood provides a minimally invasive method for biomarker discovery in cancer. Longitudinal interrogation allows monitoring or prediction of therapy response, detection of minimal residual disease or progression, and determination of prognosis. Despite inherent phenotypic heterogeneity and differences in cell surface marker expression, most CTC isolation technologies typically use positive selection. This necessitates the optimization of marker-independent CTC methods, enabling the capture of heterogenous CTCs. The aim of this report is to compare a size-dependent and a marker-dependent CTC-isolation method, using spiked esophageal cells in healthy donor blood and blood from patients diagnosed with esophageal adenocarcinoma.

### Methods

Using esophageal cancer cell lines (OE19 and OE33) spiked into blood of a healthy donor, we investigated tumor cell isolation by Parsortix post cell fixation, immunostaining and transfer to a glass slide, and benchmarked its performance against the CellSearch system. Additionally, we performed DEPArray cell sorting to infer the feasibility to select and isolate cells of interest, aiming towards downstream single-cell molecular characterization in future studies. Finally, we measured CTC prevalence by Parsortix in venous blood samples from patients with various esophageal adenocarcinoma tumor stages.

**Funding:** The author(s) received no specific funding for this work.

**Abbreviations:** APC, Allophycocyanin; AUC, Area Under the Curve; CD45, Cluster of Differentiation 45; CK, Cytokeratin; CP-5FU, Cisplatin-5-Fluoro-Uracil; CROSS, ChemoRadiotherapy for Oesophageal cancer followed by Surgery Study; CT, Computed Tomography; CTCs, Circulating Tumor Cells; cTNM, clinical TNM classification; DAPI, 4′,6-diamidino-2-phenylindole; DMEM, Dulbecco's Modified Eagle's Medium; DNA, Deoxyribonucleic Acid; EAC, Esophageal Adenocarcinoma; EC, Esophageal Cancer; ECACC, European Collection of Authenticated Cell Cultures; EMT, Epithelial-Mesenchymal Transition; EpCAM, Epithelial Cell Adhesion Molecule; FCS, Fetal Calf Serum; FDA, Food and Drug Administration; FITC, Fluorescein Isothiocyanate; FLOT, 5-Fluorouracil/Leucovorin/Oxaliplatin/docetaxel; Gy, Grey; HBS, HEPES Buffered Saline; HEPES, 4-(2-hydroxyethyl)-1-piperazineethanesulfonic acid; HR, Hazard Ratio; IF, Immunofluorescence; KCL, Potassium Chloride; LNs, Lymph nodes; MACS, Magnetic-Activated Cell Sorting; mCROSS, modified CROSS; nRCT, neoadjuvant radiochemotherapy; PBS, Phosphate-buffered Saline; PET-CT, Positron Emission Tomography-Computed Tomography; pTNM, pathologic TNM classification; RT, Radiation therapy; SCC, Squamous Cell Carcinoma; SD, Standard Deviation; WBC, White Blood Cells.

## Results

OE19 and OE33 cells were spiked in healthy donor blood and subsequently processed using CellSearch (n = 16) or Parsortix (n = 16). Upon tumor cell enrichment and enumeration, the recovery rate ranged from 76.3 ± 23.2% to 21.3 ± 9.2% for CellSearch and Parsortix, respectively. Parsortix-enriched and stained cell fractions were successfully transferred to the DEPArray instrument with preservation of cell morphology, allowing isolation of cells of interest. Finally, despite low CTC prevalence and abundance, Parsortix detected traditional CTCs (i.e. cytokeratin$^+$/CD45$^-$) in 8/29 (27.6%) of patients with esophageal adenocarcinoma, of whom 50% had early stage (I-II) disease.

## Conclusions

We refined an epitope-independent isolation workflow to study CTCs in patients with esophageal adenocarcinoma. CTC recovery using Parsortix was substantially lower compared to CellSearch when focusing on the traditional CTC phenotype with CD45-negative and cytokeratin-positive staining characteristics. Future research could determine if this method allows downstream molecular interrogation of CTCs to infer new prognostic and predictive biomarkers on a single-cell level.

## Introduction

Esophageal cancer (EC) is a lethal disease and the seventh most common cancer worldwide, accounting for 3.4% of all diagnosed cancers in 2018 [1]. Two main histological subtypes denote EC: squamous cell carcinoma (SCC) and esophageal adenocarcinoma (EAC). Because the latter is the predominant subtype in western countries, we focus on EAC as histological subtype in this study. Unfortunately, the majority of these patients are diagnosed late in the course of the disease [2, 3]. Molecular tumor profiling by means of liquid biopsies, e.g. circulating tumor DNA and circulating tumor cells (CTCs), has gained momentum, since they allow to interrogate the tumor in a minimally invasive way. Moreover, the detection and number of CTCs in blood of patients with EC is considered an independent prognostic factor [4–7].

However, CTC enumeration and analysis is hampered due to their low prevalence [8, 9], thus requiring sensitive enrichment technologies, typically using e.g. immunomagnetic- or flow cytometry-based positive selection [10–12]. CTC enrichment by positive selection relies on epithelial cell surface markers on CTCs (e.g. epithelial cell adhesion molecule (EpCAM)) [13, 14], with CellSearch being the only FDA-cleared system to date. However, metastasis-associated processes such as epithelial-to-mesenchymal transition (EMT), can result in a downregulation of epithelial characteristics of CTCs, causing EpCAM-tailored selection techniques to be inadequate to retrieve these cells, thus resulting in an underestimation of CTCs [15]. Instead, marker-independent technologies that rely on the physical properties of the cells enable CTC enrichment irrespective of CTC heterogeneity.

In this study, we evaluated a marker-independent method for CTC detection using esophageal cell lines and blood samples from EAC patients. Focusing on traditional CTCs (i.e. DAPI +/cytokeratin$^+$/CD45$^-$) this method was compared against the CellSearch system using esophageal cell lines. Finally, we performed cell-based image analysis on the DEPArray platform to compare cell morphology and phenotypic features of enriched tumor cells by Parsortix or CellSearch, as this is a prerequisite for downstream single cell isolation and molecular

characterization in future studies. Due to low CTC prevalence in blood of patients with curative disease, DEPArray analysis was only performed in esophageal cell lines.

## Materials and methods

### 1. Healthy donor spiking experiments with EAC cell lines

Thirty-six peripheral blood samples, from 8 healthy blood donors, collected in CellSave Preservative (Menarini) and Cell-free DNA BCT Streck tubes, were spiked with human Caucasian EAC cell lines OE33 (JROECL33) and OE19 (JROECL19), aiming for 200 spiked cells per donor blood sample of 9mL. The experimental design is depicted in S1 Table in S1 Text. Healthy donor blood sampling and cell culture conditions are described in S1 Text. The use of venous blood from healthy subjects was approved by the ethical committee of the Ghent University Hospital (reference number: B670201628317).

### 2. Accrual of patients with EAC for CTC enumeration

We recruited stage I—IV histologically proven EAC patients starting a new curative or palliative treatment at the Ghent University Hospital and Leuven University Hospital between September 2017 and September 2018. Before treatment, the patients underwent a full clinical work-up including a physical examination, laboratory analysis, computed tomography (CT) and/or positron emission tomography computed scan (PET-CT), a gastroscopy or endoscopic ultrasound and a baseline peripheral blood draw for CTC analysis. The inclusion criteria for patient selection and treatment schemes are described in S1 Text. Approval from the ethical committee was confirmed and written informed consent was obtained from the study patients (reference numbers: B670201628319, B670201628317).

### 3. Spiked tumor cell or patient CTC enrichment by Parsortix and CellSearch

Spiked and patient blood samples were processed between 24 and 72 hours on CellSearch and Parsortix. Parsortix procedures on Cell-free DNA BCT-collected blood samples were performed with HEPES buffered saline using PX2_PF, PX2_S99F, PX2_CT2 and PX2_H programs consequently with the 6.5 μm cassettes. Upon harvest of enriched cell fractions (by applying a reverse flow to the cassette using 1.2 mL of HBS) in 1.5 mL Protein Lobind tubes, samples were stored at 4˚C up to 2 days for downstream immunostaining, glass slide transfer and immunofluorescence read-out of CTC count (vide infra). CellSearch enrichment and enumeration procedures on CellSave-collected blood samples were performed with the CellSearch Epithelial Cell kit, as previously described [16]. After analysis, cell suspension was stored in the CellSearch cartridges at 4˚C in the dark.

### 4. Immunofluorescent staining and enumeration by IF microscopy of Parsortix™-enriched cell suspensions

Parsortix-enriched cell suspensions were centrifuged (400xg, 10 min) at room temperature. Upon removal of supernatant, cell suspensions were stained with immunofluorescent antibodies directed against cytokeratin and CD45, using Hoechst to counterstain nuclei, as described in S1 Text. Immunostained cells were transferred to a PAP-pen marked area (to prevent cell loss) on a poly-L-Lysine coated glass slide (Sigma), which was air dried at 60˚C for 15 min and stored at 4˚C in the dark, as previously described [17]. Cell enumeration was performed on a Zeiss Axio Observer Z1 Inverted Phase Contrast Fluorescence Microscope using 10x and 40x magnifications of brightfield (BF), DAPI, FITC (CK), and APC (CD45) channels. Tumor-

derived cells were defined as round-shaped events on BF having diameters ranging from 5 to 40 μm, with absence of DNA fragmentation, and positive for nuclear (DAPI), cytoplasmatic cytokeratin (FITC) staining, whilst being negative for CD45 (APC).

## 5. DEPArray Nxt

CellSearch- and Parsortix-enriched and immunostained samples were subjected to DEPArray Nxt (Silicon Biosystems, IT) image analysis to infer size-based morphological features of the enriched cell fractions. Upon identification of tumor and white blood cells, using the afore-mentioned definition, diameters of DAPI and FITC-positive events were exported from the DEPArray interface, and used as measure for outer diameter sizes of nuclei and cell membranes, respectively.

## 6. Statistical analysis

All data were analyzed using descriptive statistics. Continuous variables were summarized using measures of central tendency and variability. Categorical variables were summarized using absolute and relative frequencies. Differences between groups were assessed using a generalized linear mixed model for binary data using the logit-link with enrichment system as fixed factor and donor as random factor. For analysis of the cell-and nucleus diameters of the phenotypic characterization with DEPArray Nxt image analysis a linear mixed model was used. Corrections for simultaneous hypothesis testing were performed according to Sidak. Residual analysis by means of normal quantile plots showed that a log-transformation had to be applied to the data. All analyses were performed in S-PLUS version 8 (TIBCO Software), with a two-sided $P$-value $<0.05$ considered as statistically significant.

## Results

### 1. Tumor cell recovery efficiency using Parsortix and CellSearch using preclinical EC cell line models

To establish our CTC workflow, we spiked OE33 and OE19 esophageal cells in healthy donor blood (S1 Table in S1 Text) and compared tumor cell (TC) counts and recovery efficiencies between two CTC enrichment platforms. Representative images of identified tumor cells post Parsortix and CellSearch are presented in S1 Fig in S1 Text. On average, CellSearch resulted in a 3.5-fold higher recovery ratio in comparison to Parsortix (76.3% vs 21.3%, OR 12.2, p < 0.001) (Fig 1). No tumor cells were found in the negative controls (n = 4).

### 2. Phenotypic characterization of EC tumor cells post Parsortix and CellSearch by DEPArray Nxt image analysis

Beyond enumeration, we assessed the feasibility to interrogate phenotypic traits of CTC by transferring CellSearch and Parsortix-enriched OE19 and OE33 cell suspensions to DEPArray Nxt (n = 8). Due to low CTC prevalence in the blood of patients with curative esophageal cancer, this analysis was not feasible on patient samples and thus, only performed on esophageal cell lines. Upon cell transfer and automated DEPArray image analysis (S2 Fig in S1 Text) we observed that the cell diameters of CellSearch-enriched tumor cells were larger compared to WBCs (13.6 ± 1.8 vs 10.5 ± 1.2 μm, $p < 0.001$), which was similarly reflected in nuclei sizes (10.4 ± 1.6 vs 8.7 ± 1.2 μm, $p < 0.001$). Parsortix-enriched tumor cells showed similar findings when compared to WBCs (cell diameter: 15.6 ± 2.0 vs 11.2 ± 1.7 μm, $p < 0.001$; nucleus diameter 12.7 ± 1.7 vs 9.7 ± 1.6 μm, $p < 0.001$). Between cell lines a larger cell and nucleus diameter was observed in OE33 compared to OE19 in both enrichment platforms (both $p < 0.001$).

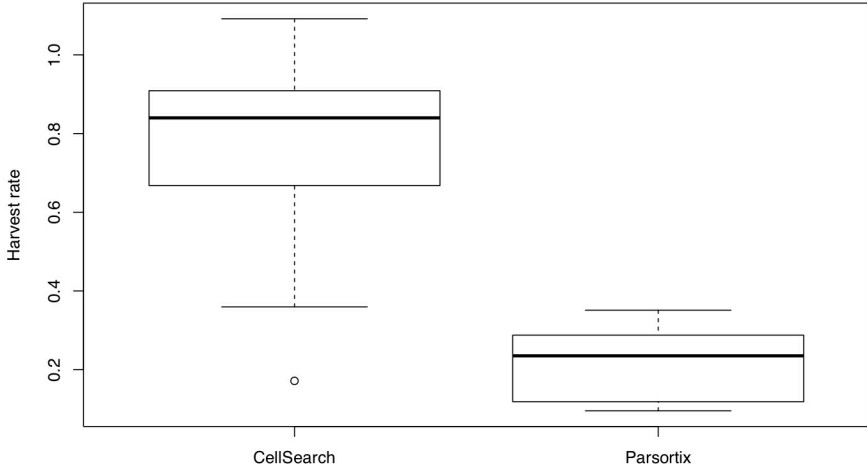

**Fig 1. Comparison of mean harvest rates of tumor cells after spike-in experiments in healthy donor blood by Parsortix and CellSearch for both OE33 and OE19.** Harvest rates. CellSearch had a harvest rate of 0.76 versus a harvest rate of 0.21 for Parsortix (p < 0.001).

Comparison of mean cell (13.6 ± 1.8 vs 15.6 ± 2.0 μm) and nucleus (10.4 ±1.6 and 12.7 ± 1.7 μm) sizes from CellSearch and Parsortix-enriched tumor cells, respectively, revealed a statistically significant difference (both *p* < 0.001) (Fig 2 and S2 Table in S1 Text). CellSearch enrichment also resulted in tumor cells and background WBC cells with a cell diameter ≤ 6.5 μm, which were absent post Parsortix.

### 3. Detection of circulating tumor cells in patients with esophageal adenocarcinoma using Parsortix

Next, we tested our established Parsortix workflow in patients with esophageal adenocarcinoma. Single peripheral blood samples were collected from EC patients (n = 29, Table 1). In 24 patients, the blood sample was collected at diagnosis prior to treatment. The other 5 patients were receiving palliative treatment for metastatic disease, with blood sampling performed between two cycles of systemic treatment (ID 31-34G, ID 42G). An overview of the patients is given in Table 1.

In total, 8/29 patients (27.6%) had ≥1 CTCs detected using Parsortix. Interestingly, 4 CTC-positive patients were at the time of blood sampling diagnosed with stage I-II disease, of whom 2 had recurrent disease, both at 8.5 months after blood sampling and curative esophagectomy (ID 6, ID 28). ID 6 died 9 months after blood sampling, ID 28 is still alive at this day (28 months of follow up). A representative image of the detected CTC in patient ID1 (tumor stage cT1bN0) is shown in Fig 3. Three of the 4 early stage CTC-positive patients had a positive nodal pathologic status. The 4 remaining CTC-positive patients were diagnosed with stage III or IV disease.

### Discussion

Whole blood taken from a patient or donor into a Streck tube could be processed immediately on the Parsortix device. The mean processing speed was 3 hours and 15 minutes which makes it suitable to use in a clinical setting. Reports on other size-based isolation methods using a filter membrane show that only small amounts of whole blood can pass through the membrane

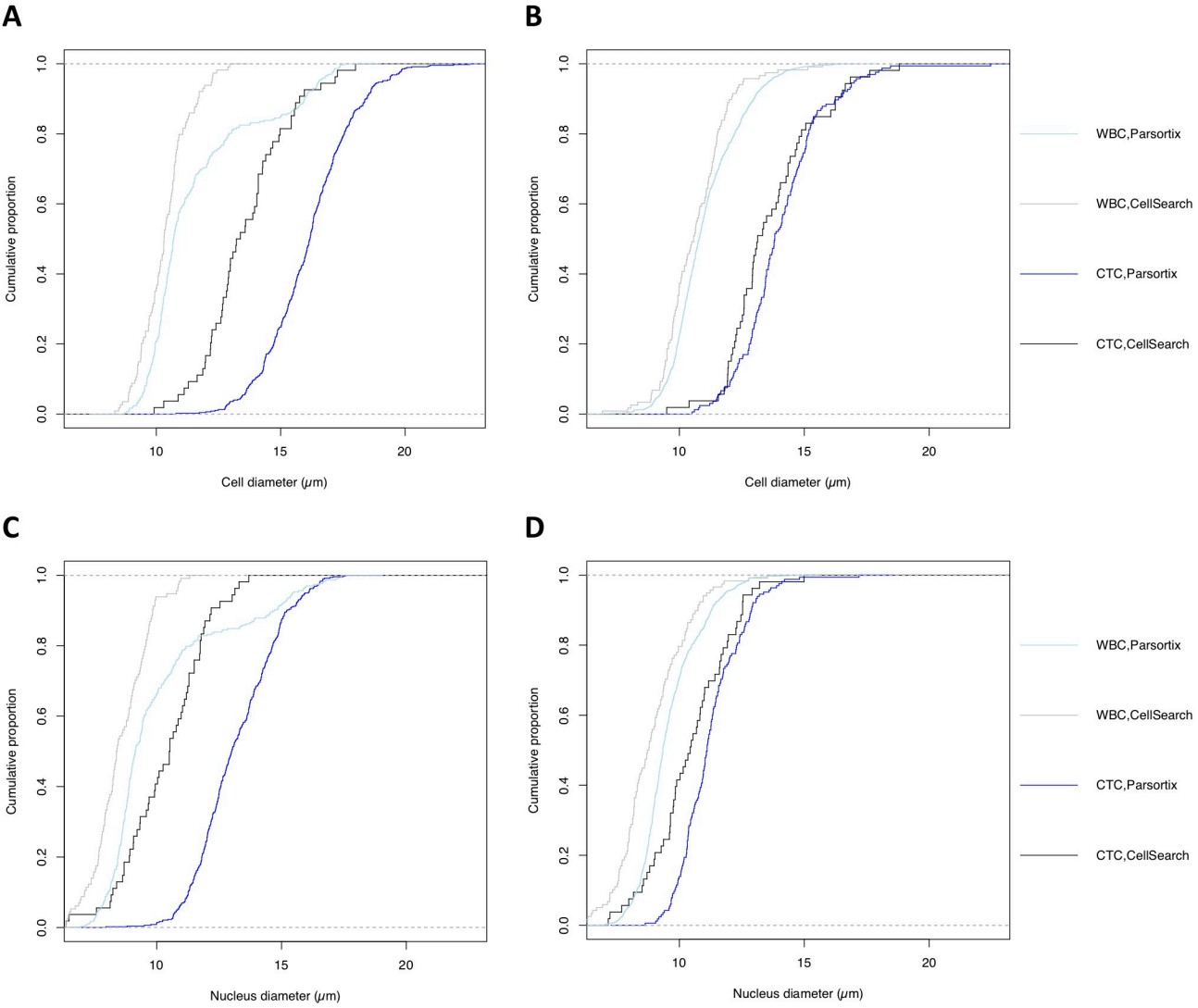

**Fig 2. Comparison of cell-and nucleus diameter per enrichment platform and cell type.** Empiric cumulative distribution curves for cell-and nucleus diameters for cell lines, WBCs and enrichment platform. A: Comparison cell diameter in OE33 cell line. B: Comparison of cell diameter in OE19 cell line. C: Comparison nucleus diameter in OE33 cell line. D: Comparison of nucleus diameter in OE19 cell line.

and that cells remain trapped in the membrane [19–21]. In contrast, Parsortix traps cells bigger than 6.5 μm in its cassette which can be flushed back and recovered for further analysis. Where CellSearch uses positive selection by selecting cells that express the EPCAM on their cell membrane, Parsortix's principle is based on the assumption that most cancer cells are much larger than peripheral hematopoietic cells such as leucocytes, as confirmed in this study. This has also been demonstrated on several cancer cell lines like PC3 and DU145 (prostate cancer cell lines) or MCF-7 (breast cancer cell lines) [21–24]. However, we may not translate this assumption to patient-derived CTCs, as some studies have found significant differences in size between cultured cells and CTCs recovered from patients [25, 26]. As expected, given the principle of Parsortix, the CTCs isolated with Parsortix were larger (15.6 ± 2.0 μm) compared to the cells isolated by CellSearch (13.6 ± 1.8 μm), suggesting that Parsortix could miss smaller CTCs.

**Table 1. Patient characteristics.**

| patient ID | age | gender | cTNM | cStage | neoadjuvant treatment | surgical resection | hist | pTNM | pstage | ctc | recurrence | death |
|---|---|---|---|---|---|---|---|---|---|---|---|---|
| 6 | 66 | m | cT1bN0 | I | N | Y | G3 | pT1bN2 | IIIA | 3 | Y | Y |
| 25 | 62 | m | cT2N0 | IIB | N | Y | G3 | pT3N2 | IIIB | 3 | N | N |
| 28 | 49 | m | cT2N0 | IIB | N | Y | G3 | pT3N1 | IIIB | 1 | Y | N |
| 1 | 60 | m | cT1bN0 | I | N | Y | G2 | pT1aN0 | IB | 1 | N | N |
| 15 | 80 | v | cT3N0 | III | N | Y | G3 | pT4aN0 | IIIB | 6 | Y | Y |
| 13 | 69 | v | cT3N1 | III | CROSS | Y | GX | ypT0N2 | IIIB | 1 | Y | Y |
| 23 | 64 | v | cT4aN1 | III | FLOT | Y | / | / | / | 1 | Y | Y |
| 32G | 55 | m | ypT2N1M+ | IVB | CROSS | Y | G2 | ypT2N1M1 | IIIA | 5 | / | Y |
| 30 | 70 | m | cT1bN0 | I | N | Y | G1 | pT1bN0 | IB | 0 | N | N |
| 21 | 62 | m | cT1bN0 | I | N | Y | G2 | pT1bN0 | IB | 0 | N | N |
| 22 | 63 | m | cT1bN0 | I | N | Y | G2 | pT1bN0 | IB | 0 | Y | N |
| 3 | 71 | m | cT1bN0 | I | N | Y | G2 | pT1bN0 | IB | 0 | Y | Y° |
| 24 | 73 | m | cT1bN0 | I | N | Y | G3 | pT1bN0 | IC | 0 | Y | Y |
| 26 | 72 | m | cT2N0 | IIB | N | Y | G1 | pT1bN0 | IB | 0 | N | N |
| 29 | 62 | m | cT2N0 | IIB | N | Y | G2 | pT2N0 | IC | 0 | N | N |
| 32 | 57 | m | cT2N0 | IIB | N | Y | G3 | pT4aN2 | IVA | 0 | Y | Y |
| 27 | 83 | v | cT3N0 | III | N | Y | G2 | pT3N0 | IIB | 0 | N | N° |
| 14 | 59 | m | cT3N1 | III | CROSS | Y | G1 | ypT3N2 | IIIB | 0 | Y | N |
| 5 | 70 | m | cT3N1 | III | mCROSS | Y | G2 | ypT1aN0 | I | 0 | Y | N |
| 10 | 71 | m | cT3N1 | III | mCROSS | Y | G2 | ypT3N2 | IIIB | 0 | Y | N |
| 11 | 58 | m | cT3N1 | III | mCROSS | Y | G3 | ypT3N2 | IIIB | 0 | Y | Y |
| 16 | 65 | m | cT3N1 | III | CROSS | Y | G3 | ypT3N2 | IIIB | 0 | Y | Y |
| 17 | 66 | m | cT3N1 | III | mCROSS | Y | G3 | ypT3N0 | II | 0 | N | N |
| 20 | 63 | m | cT3N1 | III | mCROSS | Y | / | / | / | 0 | Y | Y |
| 36G | 67 | m | cT3N2 | IVA | CROSS | Y | GX | ypT2N1 | IIIA | 0 | Y | Y |
| 31G | 51 | m | cT3N2M+ | IVB | Multiple CTs | N | G2 | / | / | 0 | / | Y |
| 33G | 51 | m | ypT3N3M+ | IVB | CP-5FU + RT 30Gy | Y | G3 | ypT3N3M1 | IVA | 0 | / | Y |
| 34G | 71 | m | ypT3N2M+ | IVB | CROSS | Y | G1 | ypT3N2M1 | IIIB | 0 | / | Y |
| 42G | 72 | m | ypT3N1M+ | IVB | CROSS | Y | G3 | ypT3N1M1 | IVB | 0 | / | Y |

cStage: clinical stage; pStage: pathologic stage [18]; m: male; f: female; Y: yes; N: no; Hist: histologic grade definitions for EAC; G1: well differentiated; G2: moderately differentiated; G3: poorly differentiated; G4: undifferentiated; GX: histologic differentiation could not be determined; y: status after completing nRCT; CTs: chemotherapies; ctc: number of CTCs found in peripheral blood using the Parsortix device; recurrence: status of disease recurrence after curative intent in EAC; death: status of cancer-related-survival after treatment;

° second primary tumor.

In our study we found that CellSearch recovered far more CTCs out of the spiked samples compared to Parsortix. This large difference could be explained by (i) the difference in immunofluorescence staining (ii) difference in cell counting technique or (iii) technical aspects of the different platforms. Where in CellSearch, the immunofluorescence staining is integrated into the fully automized process of the machine with minimal cell loss, the Parsortix procedure has included a step of manual immunofluorescence staining by the investigator with multiple washing steps and transfers to other recipients which may account for substantial cell loss [17].

Lampignano et al. compared Parsortix with CellSearch using MCF-7 breast cancer cells. They used the staining program in the Parsortix cassette provided by Angle and also counted the harvested cells after transfer to a glass slide, reporting harvest rates of 45%. The mean cell diameter size was larger compared to our esophageal cell lines (18 ± 1.7 μm versus

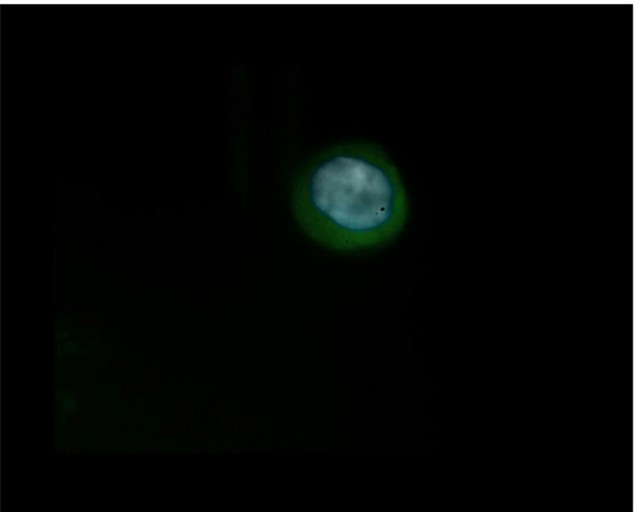

**Fig 3. Image of CTC in patient sample using Parsortix.** Immunofluorescence microscope showing a CTC in a patient sample with an irregular and larger cell nucleus (Hoechst nucleus staining (blue)) and in comparison lower volume of cell membrane (FITC-CK (green)). Staining for CD45-APC (red) was negative.

15.6 ± 2.0 μm) which can possibly explain the difference in harvest rates [27]. However, besides cell size being a critical property in Parsortix, deformability of cells under mechanical forces also plays a role in this cell isolation method [28–30]. As such, the inherent size and plasticity of different cancer cell lines could explain the different harvest rates in the literature. Xu et al. reported a harvest rate of 42.8% using a prostate cancer cell line PC3, where they spiked these cells into healthy donor blood and after recovery from Parsortix (cassette with 10 μm gap width), performed a manual staining step and transfer to a glass slide with cell count using a immunofluorescence microscope. The mean diameter of PC3 cells was 18.8 μm [17]. Hvichia et al. reported a higher harvest rate on the Parsortix ranging from 54% to 69% using different cancer cell lines (PANC-1, PC3, A 375, A549, T24 and MDA-MB-468). Interestingly, the different cell sizes reflected the difference in capture rate, the larger cancer cell line (PANC-1: mean Feret diameter 23 μm) being captured more efficiently than the smaller cancer cell lines (T24: mean Feret diameter 18 μm) [31].

In this study, we have found that 28% (8 patients out of 29) had ≥1 CTCs in their peripheral blood sample using the Parsortix for CTC isolation. Half of the CTC positive patients were diagnosed in an early stage of disease showing that hematogenous spread occurs at an early stage of tumor progression. A literature overview for detection of CTCs in esophageal cancer is presented in S3 Table in S1 Text where CTC positivity rates range between 6.4–75%. A large variability is observed in the CTC positivity rates largely due to methodological differences such as varying cut-off values and proportion of patients included with for example stage III, IV or metastatic disease. Studies including a large proportion of the latter document higher CTC-positivity rates [32–35]. Most studies use a positive selection method containing epithelial tumor markers like EPCaM [4, 6, 35–40]. Filter based methods like Screencell and Metacell select on size where after filtration immunofluorescence staining can be used like Kuvendjiska et al. did [32, 41].

In conclusion, we evaluated a marker-independent method for isolation and detection of CTCs in esophageal adenocarcinoma. Although the CellSearch outperforms Parsortix on esophageal cells, the latter showed consistent harvest rates and a cell morphology of high quality, indicating that this size-dependent technique could be used as an alternative for CellSearch when cell heterogeneity is more important than cell harvest volume. In patient blood samples

of 9 ml we found in only a few cases a low number of CTCs indicating that the enrichment method is probably not sensitive for most patients with this pathology. Apparently, the CTC abundance in patients with this tumor type is not very large and a standard blood sample is not enough to detect a significant number of CTCs.

Finally, we could differentiate phenotypic features from CTCs and WBCs isolated using the DEPArray technology, which would allow downstream molecular profiling, and warrants future research and development to optimize this workflow.

Future research should also focus on the question whether EPCAM-negative CTCs can be successfully detected using Parsortix which can open new perspectives for CTC heterogeneity analysis in esophageal adenocarcinoma. Additionally, the feasibility of molecular analysis on CTC's isolated from blood of metastatic esophageal cancer patients should be investigated.

## Supporting information

**S1 Text.**
(DOCX)

## Author Contributions

**Conceptualization:** Annouck Philippron, Lieven Depypere, Bram De Laere, Katleen De Preter, Piet Pattyn.

**Data curation:** Annouck Philippron, Lieven Depypere, Steffi Oeyen, Katleen De Preter.

**Formal analysis:** Annouck Philippron, Steffi Oeyen.

**Funding acquisition:** Piet Pattyn.

**Investigation:** Annouck Philippron, Steffi Oeyen, Katleen De Preter.

**Methodology:** Annouck Philippron, Bram De Laere, Katleen De Preter.

**Project administration:** Annouck Philippron, Piet Pattyn.

**Resources:** Annouck Philippron, Lieven Depypere, Piet Pattyn.

**Software:** Annouck Philippron.

**Supervision:** Philippe Nafteux, Katleen De Preter, Piet Pattyn.

**Validation:** Annouck Philippron.

**Visualization:** Annouck Philippron, Bram De Laere.

**Writing – original draft:** Annouck Philippron.

**Writing – review & editing:** Lieven Depypere, Bram De Laere, Charlotte Vandeputte, Philippe Nafteux, Katleen De Preter, Piet Pattyn.

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
