## [Decision Letter · Decision Letter 0]

17 Mar 2021

PONE-D-21-00260

EVALUATION OF A MARKER INDEPENDENT ISOLATION METHOD FOR CIRCULATING TUMOR CELLS IN ESOPHAGEAL ADENOCARCINOMA

PLOS ONE

Dear Dr. Philippron,

Thank you for submitting your manuscript to PLOS ONE. After careful consideration, we feel that it has merit but does not fully meet PLOS ONE’s publication criteria as it currently stands. Therefore, we invite you to submit a revised version of the manuscript that addresses the points raised during the review process.

We look forward to receiving your revised manuscript.

Kind regards,

Dominique Heymann, Ph.D.

Academic Editor

PLOS ONE

Journal Requirements:

Reviewers' comments:

Reviewer's Responses to Questions

**Comments to the Author**

1. Is the manuscript technically sound, and do the data support the conclusions?

Reviewer #1: Yes

2. Has the statistical analysis been performed appropriately and rigorously? 

Reviewer #1: Yes

3. Have the authors made all data underlying the findings in their manuscript fully available?

Reviewer #1: Yes

4. Is the manuscript presented in an intelligible fashion and written in standard English?

Reviewer #1: Yes

5. Review Comments to the Author

Reviewer #1: The manuscript submitted by Philippron et al, entitled “EVALUATION OF A MARKER INDEPENDENT ISOLATION METHOD FOR CIRCULATING TUMOR CELLS IN ESOPHAGEAL ADENOCARCINOMA” focus on the optimization of a marker-indipendent CTC enrichment which is Parsortix in particular in esophageal cancer patients.

Authors aim attention at first to compare the gold standard Cellsearch and Parsortix technologies to enrich putative CTCs (OE33 and OE19 cell lines). Authors found that CellSearch had a higher capture rate instead of Parsortix, but they chose to perform CTCs enrichment on esophageal cancer patients only by the Parsortix platform, Authors should explain better the reason for this choice.

Authors performed a morphology analysis of putative CTCs though DEPArray technology, but it is not clear why Authors did not use morphology analysis for patient samples. This is a matter of concern since authors often highlighted the potential usefulness of CTC analysis as well as molecular characterization; but they did not perform any type of this kind of analysis. More specifically Lines: 25 “the molecular characterization of circulating tumor cells (CTCs)...” and Line 39 “aiming towards downstream single-cell molecular characterization” authors stated these on the abstract objective and methods sections, but they did not perform this analysis; this kind of assumption should be addressed only in the discussion section this point is misleading.

Overall this manuscript shed a tiny light on the CTCs analysis in blood of esophageal cancer patients, but authors should pay attention to do not overstate their findings, I referred especially with DEPArray analysis which was not performed on blood patients and molecular analysis not performed as well.

6. PLOS authors have the option to publish the peer review history of their article (what does this mean?). If published, this will include your full peer review and any attached files.

Reviewer #1: No

---

## [Author Response · Author response to Decision Letter 0]

29 Mar 2021

Comments

Reviewer #1: 

1. 

Reviewer: Authors aim attention at first to compare the gold standard Cellsearch and Parsortix technologies to enrich putative CTCs (OE33 and OE19 cell lines). Authors found that CellSearch had a higher capture rate instead of Parsortix, but they chose to perform CTCs enrichment on esophageal cancer patients only by the Parsortix platform, Authors should explain better the reason for this choice. 

Authors: We chose the Parsortix platform to be evaluated, benchmarked against the Cellsearch platform. After completing the comparison between the two platforms using esophageal cell lines, we continued with Parsortix only on patient samples. We did not take extra blood samples for analysis on Cellsearch because this was not the goal set forward in this study. 

2. 

Reviewer: Authors performed a morphology analysis of putative CTCs though DEPArray technology, but it is not clear why Authors did not use morphology analysis for patient samples. This is a matter of concern since authors often highlighted the potential usefulness of CTC analysis as well as molecular characterization; but they did not perform any type of this kind of analysis. More specifically Lines: 25 “the molecular characterization of circulating tumor cells (CTCs)...” and Line 39 “aiming towards downstream single-cell molecular characterization” authors stated these on the abstract objective and methods sections, but they did not perform this analysis; this kind of assumption should be addressed only in the discussion section this point is misleading. 

Authors: We realise this can be misleading. We did not perform the same analysis with DEPArray on the patient samples because the number of CTC’s per blood tube from an esophageal cancer patient in currative setting is simply too few. Future aspects in this study are that a higher CTC prevalence can be found in metastatic patients which can make CTC analysis- including molecular characterization- possible. We will adapt all reference to ‘molecular characterization’ in the manuscript to avoid misinterpretation and add this aspect to future perspectives at the end in the section ‘Discussion’.

3. 

Reviewer: Overall this manuscript shed a tiny light on the CTCs analysis in blood of esophageal cancer patients, but authors should pay attention to do not overstate their findings, I referred especially with DEPArray analysis which was not performed on blood patients and molecular analysis not performed as well, descriptive manuscript, carefull to not overstate findings, we carefully reviewed this manuscript

Authors: We agree with the reviewer and will revise this descriptive manuscript to not overstate our findings. However, we were carfull to write a ‘descriptive’ manuscript and to not formulate statements because of the more explorative study with low number of patient samples.

---

## [Decision Letter · Decision Letter 1]

20 Apr 2021

EVALUATION OF A MARKER INDEPENDENT ISOLATION METHOD FOR CIRCULATING TUMOR CELLS IN ESOPHAGEAL ADENOCARCINOMA

PONE-D-21-00260R1

Dear Dr. Philippron,

We’re pleased to inform you that your manuscript has been judged scientifically suitable for publication and will be formally accepted for publication once it meets all outstanding technical requirements.

Kind regards,

Dominique Heymann, Ph.D.

Academic Editor

PLOS ONE

Additional Editor Comments (optional):

Reviewers' comments:

Reviewer's Responses to Questions

**Comments to the Author**

1. If the authors have adequately addressed your comments raised in a previous round of review and you feel that this manuscript is now acceptable for publication, you may indicate that here to bypass the “Comments to the Author” section, enter your conflict of interest statement in the “Confidential to Editor” section, and submit your "Accept" recommendation.

Reviewer #1: All comments have been addressed

2. Is the manuscript technically sound, and do the data support the conclusions?

Reviewer #1: Yes

3. Has the statistical analysis been performed appropriately and rigorously? 

Reviewer #1: Yes

4. Have the authors made all data underlying the findings in their manuscript fully available?

Reviewer #1: Yes

5. Is the manuscript presented in an intelligible fashion and written in standard English?

Reviewer #1: Yes

6. Review Comments to the Author

Reviewer #1: (No Response)

7. PLOS authors have the option to publish the peer review history of their article (what does this mean?). If published, this will include your full peer review and any attached files.

Reviewer #1: No

---

## [Editor Report · Acceptance letter]

29 Apr 2021

PONE-D-21-00260R1 

EVALUATION OF A MARKER INDEPENDENT ISOLATION METHOD FOR CIRCULATING TUMOR CELLS IN ESOPHAGEAL ADENOCARCINOMA 

Dear Dr. Philippron:

I'm pleased to inform you that your manuscript has been deemed suitable for publication in PLOS ONE. Congratulations! Your manuscript is now with our production department. 

Kind regards, 

on behalf of

Pr. Dominique Heymann 

Academic Editor

PLOS ONE